# Post-COVID spirometric abnormalities in workers with intermittent high-altitude exposure: A cross-sectional study in Peru

Jair Alonso Góngora-Bendezú[1], Marleyssi Valeria Martinez-López[1],
Angel David Aguinaga-Fernandez[1], Marlon Yovera-Aldana[2]*

1 Maestría en Medicina Ocupacional y Medio Ambiente, Universidad Científica del Sur, Lima, Perú,
2 Grupo de investigación de Neurociencias, Metabolismo, Efectividad Clínica y Sanitaria, Universidad Científica del Sur, Lima, Perú

* myovera@cientifica.edu.pe

## Abstract

### Introduction

Persistent pulmonary sequelae after SARS-CoV-2 infection remain a concern in workers exposed to environmental and physiological stressors such as intermittent high-altitude hypoxia. Few studies include pre-pandemic spirometry or focus on this occupational group.

### Methods

A cross-sectional study was conducted in 400 sea-level-born workers intermittently exposed to altitudes >2500 m, all with confirmed COVID-19. Only A/B-quality spirometry from 2024 was included. Sociodemographic, clinical, and occupational variables were assessed, and adjusted prevalence ratios (aPR) were estimated using Poisson regression with robust variance.

### Results

Overall, 72.2% of workers showed spirometric abnormalities (40.5% mixed, 20.8% restrictive, 11.0% obstructive). Independent predictors included obesity (aPR 1.35; 95% CI 1.19–1.53), higher Charlson index (aPR 1.49; 95% CI 1.33–1.68), ≥ 5 years of inhalant exposure (aPR 1.64; 95% CI 1.43–1.89), ≥ 7 years of intermittent high-altitude exposure (aPR 1.81; 95% CI 1.60–2.05), and severe COVID-19 (aPR 1.65; 95% CI 1.41–1.91).

### Conclusions

Over 70% of participants showed abnormal spirometry was found in post-COVID-19 workers with intermittent high-altitude exposure. Respiratory function monitoring

**Data availability statement:** All relevant data are within the paper and its Supporting Information files.

**Funding:** The author(s) received no specific funding for this work.

**Competing interests:** The authors have declared that no competing interests exist.

should be reinforced in this occupational group, especially among those with higher clinical and environmental risk factors.

## Introduction

Coronavirus disease 2019 (COVID-19), caused by the SARS-CoV-2 virus, has resulted in a substantial burden of long-term sequelae, particularly involving the respiratory system. Even in individuals without prior pulmonary conditions, post-infectious abnormalities in lung function have been frequently reported months after recovery, with clinical presentations ranging from persistent dyspnea to subclinical changes detectable only via pulmonary function testing [1]. Among these abnormalities, restrictive, obstructive, and mixed ventilatory defects have been observed, reflecting diverse underlying pathophysiological mechanisms such as interstitial inflammation, airway remodeling, and microvascular damage [2,3].

While these sequelae are increasingly documented in hospital-based and community populations, little is known about their impact in occupational groups exposed to extreme environmental conditions such as high-altitude workers [4]. Chronic or intermittent exposure to hypobaric hypoxia induces physiological adaptations, including increased pulmonary artery pressure, polycythemia, and vascular remodeling, but may also predispose susceptible individuals to maladaptive responses and impaired pulmonary recovery after infections [5,6]. Moreover, miners and other laborers at altitude frequently encounter additional hazards, including exposure to particulate matter, diesel fumes, and welding gases, all of which have been implicated in chronic airway inflammation and small airway disease [6,7].

In occupational health surveillance programs in Peru and other Andean countries, spirometry is commonly used as part of functional evaluations to assess respiratory fitness for high-altitude labor [8]. However, the COVID-19 pandemic disrupted baseline assessments and introduced a new, poorly characterized risk factor for post-infectious spirometric abnormalities [9]. Importantly, the combined effects of SARS-CoV-2 infection, metabolic risk factors such as obesity, and environmental exposures have yet to be evaluated in a population with documented normal lung function prior to the pandemic [10].

Recent evidence suggests that obesity, advanced age, and pre-existing comorbidities increase the likelihood of persistent pulmonary dysfunction post-COVID-19, even in mild cases [11]. However, few studies have explored the long-term respiratory impact of COVID-19 in high-altitude workers with pre-pandemic normal spirometry, where hypoxia and inhaled irritants may exacerbate residual inflammation or hinder tissue repair [12].

Although some longitudinal studies have evaluated respiratory sequelae after SARS-CoV-2 infection, most lacked pre-COVID pulmonary function data, limiting the ability to distinguish true post-infectious impairment from pre-existing abnormalities. Additionally, evidence in occupational groups exposed to intermittent high-altitude hypobaric conditions remains extremely scarce, despite their unique physiological vulnerabilities. To address these important gaps, this study evaluates the prevalence

and associated factors of abnormal spirometry in workers with documented normal pre-pandemic lung function and a history of COVID-19, who also experience intermittent exposure to high altitude.

## Materials and methods

### Study design and setting

We conducted a cross-sectional analytical study using data from occupational medical evaluations conducted in 2024 at a high-altitude occupational health center located in Cusco, Peru, at an elevation of 3,400 meters above sea level. The facility provides routine medical evaluations for workers from four economic sectors: mining, administration, construction, and the preservation industry. Pre-pandemic spirometry was used solely to confirm normal baseline lung function for eligibility; no longitudinal comparisons were performed, and the study design remained cross-sectional.

### Population and sampling

The study population included male and female workers residing at sea level who had a confirmed diagnosis of coronavirus disease 2019 (COVID-19) by reverse transcription polymerase chain reaction (RT-PCR) between 2020 and 2021 and underwent comprehensive annual occupational medical evaluations in 2024, including spirometry. Only spirometric tests graded as A or B in quality, according to the standards of the American Thoracic Society and European Respiratory Society (ATS/ERS), were included [13]. Participants were required to have no prior history of chronic pulmonary disease and to have had normal spirometric values before the onset of the COVID-19 pandemic.

We excluded workers currently performing tasks with high occupational risk for chronic respiratory conditions, such as those exposed to silica, asbestos, mineral dust, or industrial vapors (e.g., textile workers processing pita fiber, miners, and construction workers exposed to cement or inorganic particulates). Workers with incomplete or inconsistent medical records, as well as those who had resided at altitudes of 2,500 meters above sea level (m.a.s.l.) or higher for more than six consecutive months, were also excluded.

The sample size was estimated using the formula for a single population proportion, utilizing the OpenEpi software (version 3.01). An expected prevalence of abnormal spirometry of 32% was adopted, based on previous studies conducted in occupational populations exposed to similar altitudes in Latin America [14]. Using a 95% confidence level and a 5% margin of error, the minimum required sample size was calculated to be 334 participants. To account for potential exclusions due to invalid spirometry, incomplete data, or ineligibility, an additional 20% was added. Consequently, the final sample size was set at 400 participants, selected by simple random sampling.

### Study variables

The primary outcome was spirometric abnormality, defined as the presence of any non-normal ventilatory pattern (obstructive, restrictive, or mixed), classified according to ATS/ERS guidelines. Independent variables included age (categorized as <40, 40–50, 50–60, and >60 years), sex, body mass index (BMI), Charlson Comorbidity Index (CCI), job category, duration of employment, prior exposure to harmful agents, severity of COVID-19, and years of intermittent exposure to high altitude.

Inhalational exposure was defined based on documented contact with respiratory irritants, including silica dust, cement particulates, diesel exhaust, metal welding fumes, and other inorganic aerosols identified in occupational risk assessments. Exposure history was classified using three data sources: (1) occupational health records, (2) job-task and workplace hazard documentation, and (3) a structured self-report questionnaire completed during the medical evaluation. Years of exposure were categorized as 1–2.9, 3–4.9, and ≥5 years.

BMI was categorized according to World Health Organization (WHO) criteria: underweight (<18.5 kg/m²), normal (18.5–24.9 kg/m²), overweight (25.0–29.9 kg/m²), and obesity (≥30 kg/m²). The CCI was classified into no comorbidities

(0 points), Low burden (1–2 points), and high burden (≥3 points) comorbidity burden [15]. Occupation was categorized as white-collar (administrative roles) or blue-collar (technical, operational, supervisory, and health, safety, and environment personnel).

Employment duration and high-altitude exposure were categorized into three ranges: 3–5 years, 5–7 years, and ≥7 years. COVID-19 severity was classified as mild (outpatient management), moderate (hospitalization), or severe (intensive care unit admission).

## Spirometry procedures

Spirometry was conducted in Cusco, Peru, at an altitude of 3,400 meters above sea level, using an AMIR Espiro® spirometer. Calibration was performed daily using a 3-L syringe, and all measurements were adjusted to ambient temperature, pressure, and humidity (BTPS conditions). In addition, device calibration was verified every six months according to the manufacturer's specifications. The procedures adhered to the 2019 ATS/ERS standards including all criteria for maneuver acceptability, reproducibility, and quality control. At least three acceptable and two reproducible forced maneuvers were required (with $FEV_1$ and FVC within 150 mL), each with a forced expiratory time of at least six seconds. The measured parameters included forced vital capacity (FVC), forced expiratory volume in one second ($FEV_1$), and the $FEV_1$/FVC ratio [13].

Ventilatory patterns were interpreted according to ATS/ERS 2021 standards and reference values [16]. The following cut-off values were used:

• Obstructive pattern: $FEV_1$/FVC < 0.70

• Restrictive pattern: FVC < 80% of predicted with $FEV_1$/FVC ≥ 0.70

• Mixed pattern: $FEV_1$/FVC < 0.70 and FVC < 80% of predicted

A spirometry was considered abnormal if it met criteria for any of the above patterns. It was considered normal if $FEV_1$, FVC, and $FEV_1$/FVC were all within normal predicted values (i.e., $FEV_1$ and FVC ≥ 80% predicted, and $FEV_1$/FVC ≥ 0.70).

Only tests rated A or B in quality were accepted. Grade A spirometries fully complied with all ATS/ERS technical standards; and grade B spirometries met the reproducibility and interpretability criteria with only minor deviations that did not affect diagnostic validity.

## Data analysis

The data were entered into Microsoft Excel 365 and subsequently analyzed using Stata version 18 (StataCorp LLC, College Station, TX, USA). Data access and analysis were conducted between 1 November 2024 and 30 March 2025. Categorical variables were summarized as absolute and relative frequencies, while continuous variables were described using means and standard deviations. Bivariate analyses were performed using the chi-square or Fisher's exact test, as appropriate.

For multivariable analysis, a generalized linear model (GLM) was fitted using a Poisson distribution, log link function, and robust variance to estimate adjusted prevalence ratios (aPRs) and their 95% confidence intervals (CIs). Because the aim of the study was to identify factors associated with abnormal spirometry rather than infer causality, covariates were selected based on clinical and epidemiological relevance and on observed associations in descriptive analyses. For the multivariable analyses, all continuous predictors—such as age, body mass index (BMI), Charlson Comorbidity Index, duration of intermittent high-altitude exposure, and years of exposure to respiratory irritants—were included in the models as categorical variables, based on clinically and occupationally relevant cutoffs.. Collinearity among covariates was evaluated using the variance inflation factor (VIF), ensuring acceptable multicollinearity levels (VIF < 5).

A forest plot was generated using the coefplot command to visually display adjusted prevalence ratios and their 95% CIs.

## Ethical considerations

This study was conducted in accordance with the ethical principles outlined in the Declaration of Helsinki and national regulations for biomedical research. Approval was granted by the Institutional Committee of Ethics in Research (CIEI) of the Universidad Científica del Sur, under official document CONSTANCIA N°670-CIEI-CIENTÍFICA-2024. The research protocol was registered with the code POS-60-2024-00841. As this study involved secondary analysis of anonymized data collected during routine occupational health evaluations, individual informed consent was not required. However, all personally identifiable information was protected to ensure strict confidentiality.

## Results

### Participant selection process

In 2024, a total of 25,423 workers were registered at the occupational health center. Of these, 21,301 were excluded due to either the absence of a confirmed history of COVID-19 or lack of a normal spirometry record from the year 2020. An additional 2,103 individuals were excluded because of invalid post-COVID spirometry results, absence of intermittent high-altitude exposure, or incomplete occupational or clinical records. This resulted in a pool of 2,109 eligible participants, from which a simple random sample of 400 individuals was selected for the study (Fig 1).

Flow diagram illustrating participant selection, reasons for exclusion at each stage, and the final sample included in the analysis.

### Participant characteristics

The study population consisted mainly of middle-aged adults, with a mean age of 47 years, and men and women represented in nearly equal proportions. Approximately two thirds of the workers had excess body weight (BMI ≥ 25), including a subset who met criteria for obesity. A little under half of the participants reported at least one comorbidity, and about one in five had a high comorbidity burden (Table 1).

### COVID-19 Severity and Occupational Exposure

Regarding the clinical course of COVID-19, around two in five workers required hospitalization, and approximately one in twelve needed intensive care.

In terms of occupational profile, administrative staff constituted the white-collar subgroup, accounting for roughly one in six workers. All other positions were classified as blue-collar roles, with technical, supervisory, and environmental health and safety personnel collectively representing about one third of the workforce. Approximately one in five workers performed operator functions, corresponding to the lower-specialization tier within blue-collar occupations. Because eligibility required a documented normal pre-COVID spirometry in 2019, all participants had at least three years of employment, and roughly two thirds had more than five years in their current company. More than half of the workers had some history of exposure to respiratory irritants, with approximately one in ten reporting long-term exposure exceeding five years. Intermittent high-altitude exposure was also common, and nearly two thirds of workers had accumulated more than three years of exposure (Table 1).

### Spirometric Findings

Most workers presented post-COVID ventilatory abnormalities, with nearly three-quarters showing an altered spirometric pattern. The mixed pattern was the most frequent, followed by obstructive and restrictive types. Approximately half of the participants retained preserved $FEV_1$ values, whereas the rest exhibited mild-to-moderate reductions in expiratory flow. A substantial proportion also had reduced FVC, and more than half showed a decreased $FEV_1$/FVC ratio, indicating combined airway and volume impairments (Table 2).

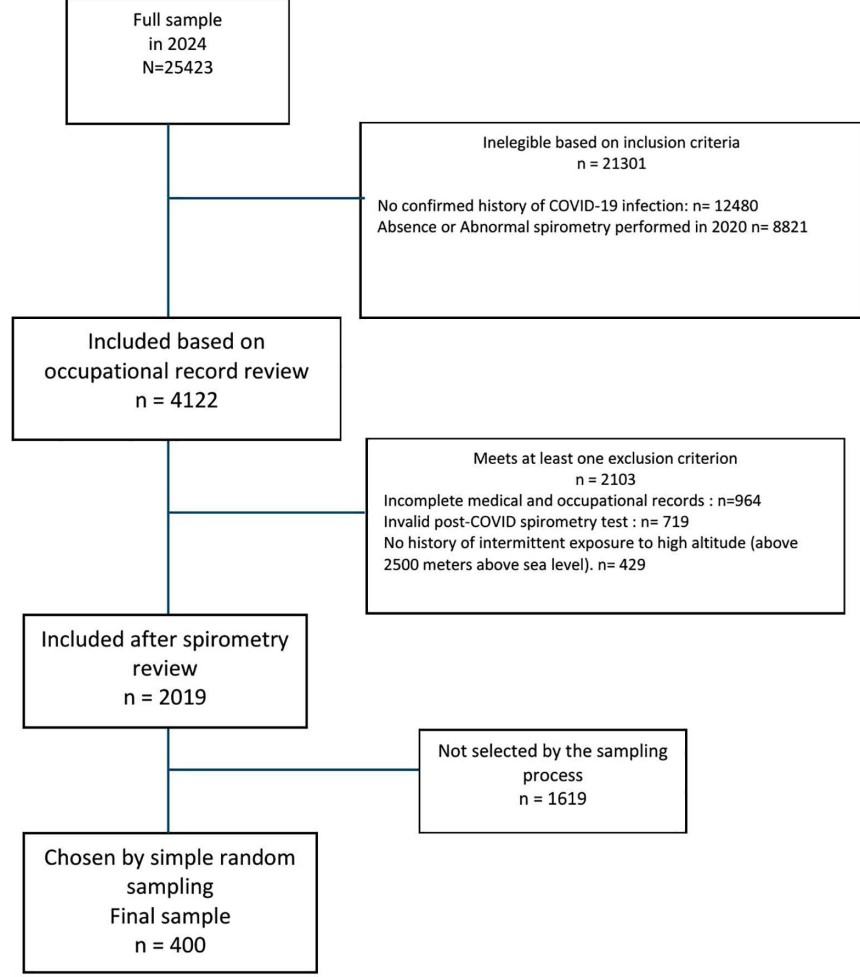

**Fig 1. Flowchart of Study Population Selection.**

## Bivariate and multivariate analysis

In bivariate analysis, abnormal spirometry was significantly associated with obesity ($p < 0.001$), Charlson Index ≥2 ($p = 0.003$), moderate-to-severe COVID-19 ($p < 0.001$), high-altitude exposure ≥7 years ($p = 0.002$), and inhalant exposure ≥5 years ($p = 0.001$) (Table 3).

These same variables remained independently associated with abnormal spirometry in the multivariable model, with the strongest associations observed for severe COVID-19, high-altitude exposure of seven years or more, and inhalational exposure exceeding five years. Neither sex, occupation, nor employment duration demonstrated meaningful associations after adjustment. Assessment of multicollinearity showed acceptable levels among all covariates, with variance inflation factors (VIFs) below 5 for every predictor included in the multivariable model (**Table 4 and Fig 2**).

Adjusted prevalence ratios (aPRs) with 95% confidence intervals were estimated using Poisson regression with robust variance. All continuous predictors were modeled as categorical variables according to clinically and occupationally relevant cutoffs. The vertical dashed line indicates the null value (aPR = 1.0).

**Table 1. Characteristics of the Study Population.**

| Category | Frequency N (%) |
|---|---|
| **Age** | |
| < 40 years | 135 (33.8) |
| 40–49 years | 100 (25.0) |
| 50–59 years | 115 (28.7) |
| 60–65 years | 50 (12.5) |
| X ± SD | 47 ± 11.7 |
| **Sex** | |
| Male | 212 (53.0) |
| Female | 188 (47.0) |
| **Body Mass Index (BMI)** | |
| Normal (< 25 kg/m²) | 134 (33.5) |
| Overweight (25.0–29.9 kg/m²) | 184 (46.0) |
| Obesity (≥ 30 kg/m²) | 82 (20.5) |
| X ± SD | 26.8 ± 4.0 |
| **Charlson Comorbidity Index** | |
| No comorbidities | 189 (47.3) |
| Low burden (Charlson Index 1 or 2) | 125 (31.2) |
| High burden (Charlson Index ≥3) | 86 (21.5) |
| **COVID-19 Severity** | |
| Outpatient management | 241 (60.3) |
| General hospitalization | 124 (31.0) |
| Intensive care | 35 (8.7) |
| **Occupation** | |
| Administrative | 69 (17.2) |
| Supervisor | 66 (16.5) |
| Environmental Health and Safety | 127 (31.8) |
| Technician | 65 (16.3) |
| Operator | 73 (18.3) |
| **Duration of Employment in Current Company** | |
| 3 - 4.9 years | 144 (36.0) |
| 5 - 6.9 years | 123 (30.7) |
| ≥ 7 years | 133 (33.2) |
| **Previous Occupational Exposure to Respiratory Irritants** | |
| No | 186 (46.5) |
| 1 - 2.9 years | 87 (21.7) |
| 3 - 4.9 years | 88 (22.0) |
| ≥ 5 years | 39 (9.7) |
| **Intermittent High-Altitude Exposure** | |
| <4 years | 165 (41.3) |
| 4 - 6.9 years | 120 (30.0) |
| ≥ 7 years | 115 (28.8) |

Continuous variables are presented as mean ± standard deviation. All continuous predictors were categorized using clinically and occupationally relevant thresholds for descriptive and multivariable analyses. Categorical variables are presented as absolute numbers and percentages

**Table 2. Spirometric Characteristics and Prevalence of Respiratory Patterns.**

| Variable | n | Prevalence (95% CI) |
|---|---|---|
| **FEV₁ Classification (mL)** | | |
| >70 | 185 | 46.2% (41.2–51.2) |
| 60–69 | 20 | 5.0% (3.1–7.6) |
| 50–59 | 60 | 15.0% (11.6–18.8) |
| 35–49 | 125 | 31.2% (26.7–36.0) |
| <35 | 10 | 2.5% (1.2–4.5) |
| **FVC (% of predicted value)** | | |
| <80% | 155 | 38.8% (33.9–43.7) |
| ≥80% | 276 | 69.0% (64.2–73.5) |
| **Tiffeneau Index (FEV₁/FVC, %)** | | |
| <70% | 206 | 51.5% (46.4–56.5) |
| ≥70% | 194 | 48.5% (43.5–53.5) |
| **Spirometric Alteration** | | |
| Normal | 111 | 27.8% (23.4–32.4) |
| Abnormal | 289 | 72.2% (67.5–76.5) |
| – Restrictive/obstructive pattern | 162 | 40.5% (35.6–45.4) |
| – Obstructive pattern only | 83 | 20.8% (16.9–25.1) |
| – Restrictive pattern only | 44 | 11.0% (8.1–14.5) |

Ventilatory patterns were classified according to ATS/ERS 2021 criteria (obstructive: $FEV_1/FVC < 0.70$; restrictive: FVC < 80% predicted with preserved ratio; mixed: both reduced). $FEV_1$ = Forced Expiratory Volume in the first second; FVC = Forced Vital Capacity;

95% CI = 95% Confidence Interval

The frequency of spirometric pattern abnormalities across clinical and occupational categories is presented in **S1 Table**, while descriptive statistics of spirometric indices ($FEV_1$, FVC, and $FEV_1$/FVC ratio) by demographic, clinical, and occupational characteristics are summarized in **S2 Table**.

## Discussion

### Main findings

This study found a considerable burden of new-onset spirometric abnormalities among high-altitude workers who had recovered from COVID-19, despite having normal lung function before the pandemic. Mixed ventilatory patterns predominated, suggesting combined airway and interstitial involvement. Factors such as excess body weight, higher comorbidity burden, more severe acute infection, many years of intermittent high-altitude exposure, and prolonged inhalant exposure were independently associated with abnormal spirometry, reflecting an interaction between individual susceptibility and demanding environmental conditions. The availability of pre-pandemic spirometry provided a clear baseline, supporting the interpretation that these abnormalities represent post-COVID functional decline rather than pre-existing disease.

### Comparison with other studies

**Abnormal spirometry prevalence.** The prevalence of post-COVID-19 spirometric abnormalities observed in our study is higher than that reported in most published cohorts, although differences across studies must be interpreted in light of follow-up duration, severity of acute infection, and environmental context. High-quality systematic reviews indicate that pulmonary function abnormalities are most frequent within the first 3–6 months after infection, with impaired diffusing capacity (DLCO) reported in 35–47% of patients and restrictive defects in approximately 8–13%, while obstructive

**Table 3. Prevalence of spirometric abnormalities according to demographic, clinical, and occupational characteristics.**

| Variable | Abnormal Spirometry | Normal Spirometry | p-Value |
|---|---|---|---|
| **Sex** | | | |
| Female | 159 (75.0) | 53 (25.0) | 0.192 [a] |
| Male | 130 (69.1) | 58 (30.9) | |
| **Age** | | | |
| < 40 years | 102 (75.6) | 33 (24.4) | 0.213 [a] |
| 40–49 years | 71 (71.0) | 29 (29.0) | |
| 50–59 years | 76 (68.0) | 39 (33.9) | |
| 60–65 years | 40 (80.0) | 10 (20.0) | |
| **Body Mass Index (BMI)** | | | |
| Normal (<25 kg/m²) | 92 (68.7) | 42 (31.3) | <0.001[a] |
| Overweight (25–29.9 kg/m²) | 115 (62.5) | 69 (37.5) | |
| Obesity (≥30 kg/m²) | 82 (100.0) | 0 (0.0) | |
| **Charlson Comorbidity Index** | | | |
| No comorbidities | 119 (63.0) | 70 (37.0) | < 0.001 [a] |
| Low burden (Charlson Index 1 or 2) | 84 (67.2) | 41 (32.8) | |
| High burden (Charlson Index ≥3) | 86 (100.0) | 0 (0.0) | |
| **COVID-19 Severity** | | | |
| Outpatient management | 172 (71.3) | 69 (28.6) | < 0.001 [a] |
| General hospitalization | 82 (66.1) | 42 (33.9) | |
| Intensive care | 35 (100.0) | 0 (0.0) | |
| **Occupation** | | | |
| Administrative | 48 (69.6) | 21 (30.4) | 0.9280 [a] |
| Supervisor | 44 (68.7) | 20 (31.3) | |
| Environmental, health and safety | 101 (73.2) | 37 (26.8) | |
| Technician | 47 (74.6) | 16 (25.4) | |
| Operator | 49 (74.2) | 17 (25.7) | |
| **Duration of employment** | | | |
| 3–4.9 years | 106 (73.6) | 38 (26.4) | 0.762 [b] |
| 5–7.9 years | 90 (73.2) | 33 (26.8) | |
| ≥7 years | 93 (69.9) | 40 (30.1) | |
| **Previous Occupational Exposure to Respiratory Irritants** | | | |
| None | 121 (65.1) | 65 (34.9) | < 0.001 [a] |
| 1–2.9 years | 65 (74.7) | 22 (25.3) | |
| 3–4.9 years | 64 (72.7) | 24 (27.3) | |
| ≥5 years | 39 (100.0) | 0 (0.0) | |
| **Intermittent High-Altitude Exposure** | | | |
| <4 years | 94 (56.9) | 71 (43.1) | < 0.001 [a] |
| 4–6.9 years | 80 (66.7) | 40 (33.3) | |
| ≥7 years | 115 (100.0) | 0 (0.0) | |

p-values were obtained using the chi-square test (a) or the Mann–Whitney U test (b), according to variable characteristics. Categorical variables are presented as absolute and relative frequencies

**Table 4. Prevalence ratios for abnormal spirometry according to demographic and clinical factors in workers with intermittent high-altitude exposure.**

| Variable | Crude PR (95% CI) | p-value | Adjusted PR (95% CI) | p-value |
|---|---|---|---|---|
| **Sex** | | | | |
| Female | Reference | — | Reference | — |
| Male | 1.08 (0.95–1.22) | 0.197 | 1.03 (0.93–1.15) | 0.462 |
| **Age** | | | | |
| < 40 years | Reference | — | Reference | — |
| 40–49 years | 0.93 (0.80–1.10) | 0.440 | 0.94 (0.82–1.09) | 0.425 |
| 50–59 years | 0.87 (0.74–1.03) | 0.106 | 0.82 (0.71–0.94) | 0.718 |
| 60–65 years | 1.05 (0.89–1.25) | 0.507 | 1.21 (1.02–1.42) | 0.022 |
| **Body Mass Index (BMI)** | | | | |
| Normal (<25 kg/m²) | Reference | — | Reference | — |
| Overweight (25–29.9 kg/m²) | 0.91 (0.77–1.06) | 0.250 | 0.92 (0.81–1.04) | 0.191 |
| Obesity (≥30 kg/m²) | 1.45 (1.29–1.63) | <0.001 | 1.35 (1.19–1.53) | <0.001 |
| **Charlson Comorbidity Index** | | | | |
| No comorbidities | Reference | — | Reference | — |
| Low burden (Charlson Index 1 or 2) | 1.06 (0.91–1.26) | 0.437 | 0.98 (0.86–1.11) | 0.733 |
| High burden (Charlson Index ≥3) | 1.59 (1.42–1.77) | <0.001 | 1.49 (1.33–1.68) | <0.001 |
| **COVID-19 Severity** | | | | |
| Outpatient management | Reference | — | Reference | — |
| General hospitalization | 0.92 (0.79–1.08) | 0.317 | 0.95 (0.84–1.07) | 0.395 |
| Intensive care | 1.40 (1.29–1.51) | <0.001 | 1.65 (1.41–1.91) | <0.001 |
| **Occupation** | | | | |
| Administrative | Reference | | | |
| Supervisor | 0.99 (0.79–1.24) | 0.919 | | |
| Environmental, health and safety | 1.05 (0.87–1.26) | 0.593 | | |
| Technician | 1.07 (0.87–1.33) | 0.519 | | |
| Operator | 1.07 (0.86–1.31) | 0.546 | | |
| **Duration of employment** | | | | |
| 3–4.9 years | Reference | — | | |
| 5–6.9 years | 0.99 (0.86–1.15) | 0.935 | | |
| ≥7 years | 0.95 (0.82–1.11) | 0.498 | | |
| **Previous Occupational Exposure to Respiratory Irritants** | | | | |
| None | Reference | — | Reference | — |
| 1–2.9 years | 1.14 (0.97–1.35) | 0.093 | 1.18 (1.04–1.35) | 0.013 |
| 3–4.9 years | 1.12 (0.94–1.32) | 0.188 | 1.08 (0.93–1.25) | 0.296 |
| ≥5 years | 1.53 (1.38–1.71) | <0.001 | 1.64 (1.43–1.89) | <0.001 |
| **Intermittent High-Altitude Exposure** | | | | |
| <4 years | Reference | — | Reference | — |
| 4–6.9 years | 1.17 (0.97–1.40) | 0.093 | 1.21 (1.04–1.40) | 0.013 |
| ≥7 years | 1.75 (1.53–2.00) | <0.001 | 1.81 (1.60–2.05) | <0.001 |

PR = Prevalence Ratio; CI = Confidence Interval Adjusted models were estimated using Poisson regression with robust variance. Occupational variables (job category and duration of employment) were excluded from the multivariable model because they showed no meaningful association with abnormal spirometry and lacked theoretical relevance as determinants of post-COVID lung function in this context.

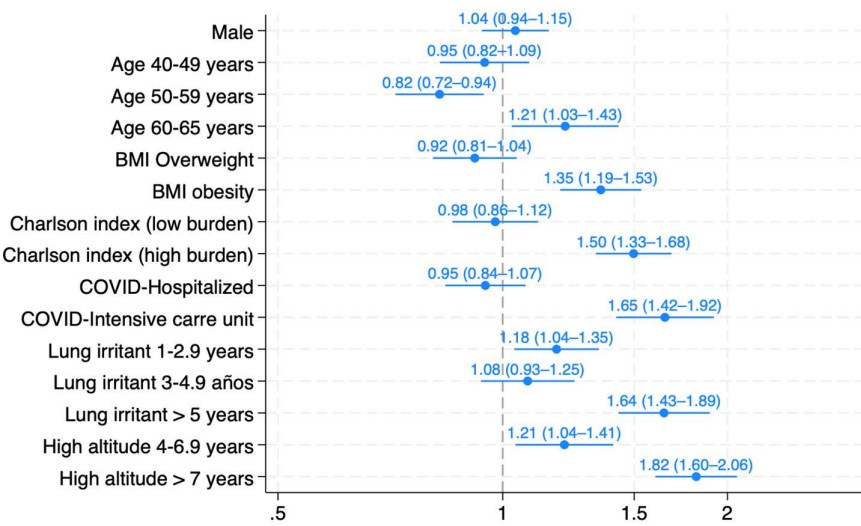

**Fig 2. Forest plot of adjusted prevalence ratios for abnormal spirometry.**

patterns remain uncommon (<10%) [17,18]. By 12 months, these abnormalities tend to improve, yet a relevant proportion of patients continue to show functional impairment, with DLCO reduction persisting in about 31% and restrictive patterns in roughly 5% [17,19]. Longer follow-up studies further demonstrate that a subset of individuals—particularly those with moderate to severe disease—may continue to exhibit reduced lung volumes or exertional limitations even 24 months post-infection [20].

Most of these data come from sea-level populations, whereas evidence from altitude-exposed groups remains limited. Notably, the multicenter FIRCOV study across Latin America found a lower prevalence of restrictive spirometry (32%) among COVID-19 survivors living at moderate to high altitudes, along with better performance on the 6-minute walk test despite lower resting and exertional oxygen saturation [21].

Taken together, current evidence shows that post-COVID spirometric abnormalities vary with clinical severity, recovery time, and environmental conditions. In our population, intermittent hypobaric exposure—rather than permanent high-altitude residence—combined with occupational inhalational risks may heighten pulmonary susceptibility and contribute to the predominance of mixed ventilatory patterns. These findings reinforce that post-COVID respiratory sequelae are context-dependent and that altitude-exposure history is essential for interpreting lung function in intermittently exposed workers.

**Risk factors for post-COVID abnormal spirometry.** In studies conducted primarily at sea level, post-COVID spirometric abnormalities has been linked to demographic characteristics (older age and male sex), clinical severity (hospitalization, oxygen therapy, or non-invasive ventilation), and comorbidities such as hypertension, diabetes, cardiovascular disease, and smoking history [22–25]. Imaging abnormalities—including consolidation, fibrosis, and ground-glass opacities—have also been associated with functional decline [25,26]. These risk factors differ from those present in our population, where intermittent high-altitude exposure and occupational inhalational risks are additional potential modifiers. This distinction highlights the relevance of evaluating post-COVID lung function within altitude- and occupation-specific contexts.

## Interpretation according to occupational and clinical factors

The predominance of mixed ventilatory patterns in this cohort suggests involvement of both airway and interstitial components. Current evidence indicates that post-COVID inflammatory or fibrotic changes may interact with occupational

exposures—such as silica, welding fumes, or diesel particulates—to amplify oxidative stress and small-airway remodeling [27–29]. Intermittent hypobaric hypoxia may further contribute by reducing diffusing capacity, impairing epithelial repair, and altering pulmonary vascular tone [30]. Although direct evidence in intermittently exposed high-altitude workers is lacking, these mechanisms provide a plausible integrated framework consistent with our findings.

Several clinical and demographic factors have been associated with abnormal post-COVID spirometry in prior studies, and many align with our results. Older age and elevated BMI have been linked to restrictive or mixed patterns due to age-related reductions in thoracic compliance and altered chest-wall mechanic [31]. Greater acute COVID-19 severity—including hospitalization, desaturation, radiographic abnormalities, and ICU admission—has repeatedly predicted persistent pulmonary dysfunction. Higher comorbidity burden has also been associated with long-term lung functional impairment [32]. In our study, longer cumulative exposure to intermittent high altitude (≥7 years) and prolonged exposure to respiratory irritants (≥5 years) were independently associated with abnormal spirometry. Previous research in Andean miners and other altitude-exposed workers documents chronic mountain sickness, hypoxemia-related remodeling, and impaired ventilatory reserve, suggesting that long-term hypoxia may reduce pulmonary resilience to additional insults such as SARS-CoV-2 [33–35].

Although males showed a slightly higher proportion of abnormal spirometry in unadjusted analyses, this difference was not retained after adjustment. Sex-related variations in baseline pulmonary physiology—such as higher absolute $FEV_1$ and FVC values typically observed in males—have been documented in other populations and may partially explain this pattern. [36] However, given the cross-sectional design and the overlap in occupational roles and exposure profiles between male and female workers, these differences should be interpreted cautiously.

## Strengths and limitations

This study has several strengths. First, the availability of pre-pandemic spirometry for all included participants enables robust comparison and supports a temporal link between COVID-19 and observed abnormalities. Second, the inclusion of multiple clinically and occupationally relevant variables enhances the external validity of the findings for similar worker populations in the Andean and Himalayan regions. Third, the use of a standardized spirometry protocol following ATS/ERS 2019 guidelines, and reference values adjusted to South American populations, ensures technical accuracy and reproducibility [13].

However, some limitations must be acknowledged. The cross-sectional design precludes evaluation of the evolution of lung function over time or assessment of reversibility. Diffusing capacity for carbon monoxide (DLCO) and high-resolution computed tomography (HRCT), which would have allowed differentiation between restrictive and diffusion-related impairments and better characterization of underlying mechanisms such as fibrosis or vascular abnormalities, were not performed. Additionally, the potential for a healthy worker effect must be considered: workers with more severe post-COVID sequelae or functional limitations may have exited the workforce before evaluation, potentially leading to underestimation of impairment. Because the study was conducted at sea level, post-exertional desaturation under actual high-altitude conditions could not be assessed. Although the sex distribution in the sample was balanced, the study did not evaluate sex-specific differences in occupational roles or exposure patterns, which may influence interpretation of spirometric outcomes. Finally, reliance on occupational health records for some exposure variables may introduce reporting bias.

## Recommendations for future research

Future studies should address the gaps identified in this investigation. Longitudinal research is needed to evaluate the progression, persistence, or resolution of post-COVID spirometric abnormalities over time, particularly among workers with prolonged high-altitude exposure or cumulative inhalant contact. Comprehensive pulmonary function testing—including DLCO, lung volumes, and high-resolution imaging—will be essential to differentiate restrictive from diffusion-related

impairments and to better characterize underlying mechanisms such as fibrosis or vascular injury. Evaluating exercise capacity, gas-exchange efficiency, and altitude-specific responses—such as post-exertional desaturation and hypoxia-challenge performance—may help clarify functional limitations that spirometry alone cannot detect. Molecular or biomarker-based studies exploring inflammatory pathways or individual susceptibility to lung injury in hypoxic environments could further refine pathophysiologic understanding.

Interventional studies should assess the effectiveness of targeted strategies, including pulmonary rehabilitation, ergonomic task modification, and exposure-reduction interventions, in restoring work capacity and preventing further decline. Health-economic evaluations are also needed to determine the feasibility and optimal design of surveillance programs in remote or resource-limited occupational settings. Finally, future research should examine potential sex-specific differences in pulmonary response to both COVID-19 and intermittent hypobaric hypoxia, ideally through stratified analyses within well-designed cohort studies.

### Public health and occupational medicine implications

These findings have important implications for occupational health surveillance in high-altitude environments. The high prevalence of subclinical spirometric abnormalities, even among workers without persistent symptoms, suggests that current return-to-work protocols—often based primarily on symptom resolution—may not adequately detect abnormal spirometry. Occupational medicine programs should consider incorporating post-COVID spirometric assessments into functional evaluations for workers exposed to extreme environments such as high altitude, particularly for those with identifiable risk factors including obesity, comorbidities, or prior moderate-to-severe COVID-19 [37].

Baseline spirometry during pre-placement evaluations is also essential, as it enables longitudinal comparison and early detection of functional decline. In this study, the availability of documented normal pre-pandemic lung function strengthened causal inference and demonstrated the value of systematic occupational health records in epidemiological assessment. [38,39].

Implementation feasibility must also be considered. In remote or resource-limited high-altitude worksites—such as mining operations—periodic full spirometry may not be practical. A stepwise surveillance approach may therefore be more appropriate: (1) routine symptom screening; (2) portable or simplified spirometry for field assessments; and (3) referral to fixed facilities for comprehensive pulmonary evaluation when abnormalities are detected. Worksite interventions, including improved ventilation systems, targeted respiratory protection, and scheduled health monitoring for workers exposed to inhalational hazards, may further mitigate risk [40].

Given the extent of high-Andean occupational activity in Peru, these findings support the development of national guidelines on occupational exposure to chronic intermittent hypobaria at high altitude, informed by emerging evidence.

At the occupational level, physicians should evaluate the relevance of incorporating functional assessments tailored to the specific demands and exposure risks of each work sector. As part of health surveillance, they should also ensure that workers with suspected pathology undergo both clinical and occupational evaluations, and that findings from these assessments are correlated to strengthen medical decision-making and the overall effectiveness of health monitoring.

### Conclusions

Our study demonstrates a high prevalence of post-COVID-19 spirometric abnormalities (>70%) among high-altitude workers, despite previously normal lung function. Obesity, higher comorbidity burden, severe acute infection, and prolonged exposure to inhalants were identified as independent predictors, and the predominance of mixed ventilatory patterns suggests combined viral and environmental effects. These findings underscore the need to strengthen respiratory surveillance in high-altitude occupational settings and support integrating feasible spirometry-based monitoring strategies into post-COVID evaluations, particularly in resource-limited environments.

## Supporting information

**S1 Table. Frequency of Spirometric Patterns by Clinical and Occupational Characteristics.**
(DOCX)

**S2 Table. Spirometric Indices (FEV$_1$, FVC, and FEV$_1$/FVC Ratio) by Demographic, Clinical, and Occupational Characteristics".**
(DOCX)

**S1 Dataset. Dataset Spirometry v2.**
(XLSX)

## Author contributions

**Conceptualization:** Jair Alonso Góngora-Bendezú, Marlon Yovera-Aldana.

**Data curation:** Jair Alonso Góngora-Bendezú, Marlon Yovera-Aldana.

**Formal analysis:** Jair Alonso Góngora-Bendezú, Marlon Yovera-Aldana.

**Investigation:** Jair Alonso Góngora-Bendezú, Marlon Yovera-Aldana.

**Methodology:** Jair Alonso Góngora-Bendezú, Marlon Yovera-Aldana.

**Project administration:** Jair Alonso Góngora-Bendezú.

**Resources:** Jair Alonso Góngora-Bendezú.

**Supervision:** Jair Alonso Góngora-Bendezú, Marleyssi Valeria Martinez-López, Angel David Aguinaga-Fernandez, Marlon Yovera-Aldana.

**Validation:** Jair Alonso Góngora-Bendezú, Marleyssi Valeria Martinez-López, Angel David Aguinaga-Fernandez, Marlon Yovera-Aldana.

**Visualization:** Jair Alonso Góngora-Bendezú.

**Writing – original draft:** Jair Alonso Góngora-Bendezú, Marlon Yovera-Aldana.

**Writing – review & editing:** Jair Alonso Góngora-Bendezú, Marleyssi Valeria Martinez-López, Angel David Aguinaga-Fernandez, Marlon Yovera-Aldana.

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
