## [Decision Letter · Decision Letter 0]

23 Oct 2025

PONE-D-25-37564Post-COVID Spirometric Abnormalities in Workers with Intermittent High-Altitude Exposure: A Cross-Sectional Study in PeruPLOS ONE

Dear Dr. Yovera-Aldana,

Thank you for submitting your manuscript to PLOS ONE. After careful consideration, we feel that it has merit but does not fully meet PLOS ONE’s publication criteria as it currently stands. Therefore, we invite you to submit a revised version of the manuscript that addresses the points raised during the review process.

Please submit your revised manuscript by Dec 07 2025 11:59PM. If you will need more time than this to complete your revisions, please reply to this message or contact the journal office at plosone@plos.org. Please include the following items when submitting your revised manuscript:

We look forward to receiving your revised manuscript.

Kind regards,

Kuryan George

Academic Editor

PLOS ONE

Journal Requirements:

Reviewers' comments:

Reviewer's Responses to Questions

**Comments to the Author**

1. Is the manuscript technically sound, and do the data support the conclusions?

Reviewer #1: Partly

2. Has the statistical analysis been performed appropriately and rigorously? 

Reviewer #1: Yes

3. Have the authors made all data underlying the findings in their manuscript fully available?

Reviewer #1: Yes

4. Is the manuscript presented in an intelligible fashion and written in standard English?

Reviewer #1: Yes

5. Review Comments to the Author

Reviewer #1: This manuscript addresses an important and timely topic: the prevalence and determinants of pulmonary function impairment among high-altitude workers following COVID-19. The study is clinically relevant, occupationally significant, and well-written. The combination of pre-pandemic spirometry, occupational exposure assessment, and post-COVID evaluation makes this work unique. The manuscript is generally well-structured, and the findings have clear implications for worker health policies.

That said, the manuscript would benefit from conciseness in results, streamlined discussion flow, and clearer emphasis on novelty. Some methodological and interpretative issues require clarification.

Major Comments

1. Novelty and Contribution

The study’s main strength is the inclusion of pre-pandemic spirometry, allowing a before–after perspective rarely available in post-COVID occupational studies.

However, the novelty is somewhat diluted in the Introduction and Discussion. The manuscript should emphasize more clearly:

A. That most prior studies lacked pre-COVID baselines.

B. That very few have studied intermittent high-altitude exposure.

Recommendation: Strengthen these points in the Introduction (last paragraph) and Discussion (main findings + conclusion).

2. Study Design and Methods

The cross-sectional design is acknowledged, but since pre-COVID spirometry exists, the study has elements of a retrospective cohort. Clarify this framing: Was the analysis truly cross-sectional, or longitudinal with before–after comparison?

Spirometry interpretation: It is not fully clear whether ATS/ERS standards were followed for quality control and reproducibility. Please clarify in Methods.

Occupational exposure assessment: More detail is needed on how “inhalational exposure” was defined (e.g., type of dusts/fumes, self-reported vs. workplace records). Right now it reads too broadly.

3. Results

Well presented, but overly detailed in the text. Many percentages are repeated verbatim from tables. This reduces readability.

Suggestion: Condense textual results to highlight only key findings, and let tables carry the detailed breakdown.

Example: Instead of listing every FEV₁ group, summarize broadly (“approximately half retained preserved function, while one-third showed moderate impairment”).

This would shorten the manuscript and improve flow.

4. Discussion

Comparison with other studies is comprehensive but reads like a series of disjointed summaries. Transitions between sea-level and high-altitude comparisons could be smoothed for narrative coherence.

Interpretation section is insightful but somewhat verbose. The mechanistic explanations (airway, interstitial, occupational toxins, hypoxia) could be merged into one integrated paragraph. Similarly, risk factor discussion could be streamlined.

At times, speculative mechanisms (e.g., immune dysregulation, environmental synergy) are presented without sufficient qualification. Consider framing these more cautiously.

The finding of higher impairment among males compared to females is intriguing but underdeveloped. The explanation (longer altitude exposure, smoking, occupational tasks) is plausible but speculative. Consider reframing more cautiously and noting that sex-specific physiological adaptations at altitude may also contribute.

5. Public Health & Policy Implications

Strong and well-articulated. The suggestion for incorporating spirometry into routine surveillance is practical and impactful.

Recommendation: Move this section closer to the Conclusion for emphasis, rather than embedding it between Interpretation and Strengths/Limitations.

The occupational recommendations are strong, but the authors could be more explicit about feasibility in low-resource settings (e.g., periodic spirometry may be unrealistic in remote mines). Consider suggesting a stepwise approach (symptom screening, portable spirometry, referral pathways).

6. Limitations

Appropriate and balanced. However, a limitation worth adding: potential healthy worker effect bias (workers with severe disease or disability may have already exited the workforce, leading to underestimation of true burden). Suggest also noting lack of DLCO or HRCT as a limitation in differentiating restriction vs. diffusion impairment.

7. Conclusion

Clear but could be more concise and impactful. Currently too wordy. Aim for a single, powerful paragraph emphasizing:

High prevalence (>70%)

Independent risk factors

Implications for surveillance and policy

Minor Comments

1. Abstract: Too dense. Consider cutting 30–40 words, focusing on the main result (72.2% prevalence, key predictors).

2. Terminology: Ensure consistent use of spirometric abnormality vs. pulmonary impairment vs. functional abnormality.

3. Numbers consistency: Minor discrepancies (72.2% vs. 72.3%) should be unified.

4. References: Overall strong, but some older references could be replaced with more recent 2023–2025 updates if available.

5. Statistical Reporting

In logistic regression, clarify whether multicollinearity was checked. Some predictors (BMI, comorbidities, duration at altitude) may be correlated.

State explicitly whether continuous predictors were analyzed as continuous or categorized.

6. PLOS authors have the option to publish the peer review history of their article (what does this mean?). If published, this will include your full peer review and any attached files.

Reviewer #1: No

---

## [Author Response · Author response to Decision Letter 1]

11 Dec 2025

Response to Reviewers

Manuscript ID:

Title: Post-COVID Spirometric Abnormalities in Workers with Intermittent High-Altitude Exposure: A Cross-Sectional Study in Peru

Dear Editor and Reviewers,

We sincerely thank you for the thorough and constructive feedback provided on our manuscript. Each comment has been carefully considered and addressed to improve clarity, scientific rigor, and full compliance with PLOS ONE editorial requirements. The revised version includes a substantially strengthened Discussion section, an expanded and fully detailed Ethics Statement, a more informative Abstract, and smoother transitions throughout the manuscript to enhance narrative coherence.

Below, we present a point-by-point response to all editorial and reviewer observations, along with a concise description of the corresponding revisions implemented in the manuscript. All modifications have been incorporated into both the tracked-changes version and the clean version submitted.

We are grateful to the reviewers and the editorial team for their insightful comments, which have significantly enhanced the quality, clarity, and robustness of our work.

Sincerely,

Marlon Yovera-Aldana

EDITORIAL REQUIREMENTS

Response: We appreciate this reminder. The manuscript has been revised to fully comply with PLOS ONE formatting requirements. Specifically: (1) the main text now follows the structure and stylistic conventions indicated in the PLOS ONE sample templates; (2) the title page, author list, and affiliations have been reformatted according to journal specifications; and (3) all files—main manuscript, figures, and supplementary materials—have been renamed following PLOS ONE’s file-naming guidelines. We have reviewed all submission components to ensure consistency with the journal’s style standards.

We appreciate the request for clarification. We confirm that the manuscript already incorporates all required ethical details regarding consent and data handling. Specifically, the Ethics Statement explains that:

The study used secondary, anonymized data obtained from routine occupational health evaluations.

No identifiable personal information was accessible to the research team.

The Institutional Committee of Ethics in Research (CIEI) of Universidad Científica del Sur reviewed and approved the study (CONSTANCIA N°670-CIEI-CIENTÍFICA-2024; protocol POS-60-2024-00841).

The CIEI waived the requirement for individual informed consent, in accordance with national regulations, because all data were fully anonymized prior to analysis and the study posed minimal risk.

No minors were included.

We appreciate this clarification. We have now added full captions for all Supporting Information files (S1 Table and S2 Table) at the end of the manuscript, following PLOS ONE formatting guidelines. In addition, all in-text citations referring to these files have been updated to ensure consistency with the final Supporting Information labeling

Reviewer #1:

1. Novelty and Contribution

The study’s main strength is the inclusion of pre-pandemic spirometry, allowing a before–after perspective rarely available in post-COVID occupational studies.

However, the novelty is somewhat diluted in the Introduction and Discussion. The manuscript should emphasize more clearly:

A. That most prior studies lacked pre-COVID baselines.

B. That very few have studied intermittent high-altitude exposure.

Recommendation: Strengthen these points in the Introduction (last paragraph) and Discussion (main findings + conclusion).

We appreciate the reviewer’s observation. We have strengthened the manuscript to more clearly highlight the two main novel aspects of the study:

(1) the availability of pre-pandemic spirometry, which is uncommon in post-COVID research and enables a true before–after comparison, and

(2) the limited evidence on post-COVID pulmonary function among workers with intermittent high-altitude exposure.

These points have been explicitly emphasized in the Introduction (final paragraph), the Discussion (Main Findings), and the Conclusion.

2. Study Design and Methods

The cross-sectional design is acknowledged, but since pre-COVID spirometry exists, the study has elements of a retrospective cohort. Clarify this framing: Was the analysis truly cross-sectional, or longitudinal with before–after comparison?

Spirometry interpretation: It is not fully clear whether ATS/ERS standards were followed for quality control and reproducibility. Please clarify in Methods.

Occupational exposure assessment: More detail is needed on how “inhalational exposure” was defined (e.g., type of dusts/fumes, self-reported vs. workplace records). Right now it reads too broadly.

We appreciate the reviewer’s request for clarification. Although pre-pandemic spirometry was available, the analysis was conducted as a cross-sectional study because no longitudinal change or within-subject comparisons were modeled. The pre-COVID spirometry was used exclusively to confirm normal lung function at baseline as part of the eligibility criteria. This clarification has been added to the Study Design subsection.

We have also expanded the Methods to explicitly state that spirometry procedures followed ATS/ERS 2019 standards for acceptability, reproducibility, and quality grading.

Finally, we have detailed how inhalational exposure was defined, including the specific types of occupational agents considered and the data sources used (occupational health records, job-task documentation, and structured self-report).

3. Results

Well presented, but overly detailed in the text. Many percentages are repeated verbatim from tables. This reduces readability.

Suggestion: Condense textual results to highlight only key findings, and let tables carry the detailed breakdown.

Example: Instead of listing every FEV₁ group, summarize broadly (“approximately half retained preserved function, while one-third showed moderate impairment”)

This would shorten the manuscript and improve flow.

We thank the reviewer for this recommendation. The Results section has been substantially revised to eliminate redundant numerical descriptions and improve narrative flow. Textual summaries corresponding to Tables 1, 2, 3, and 4 were rewritten to highlight only the key patterns and associations, avoiding repetition of percentages or category-level distributions already displayed in the tables. Detailed values now remain exclusively within the tables.

Changes to the manuscript:

Table 1 (Participant Characteristics): The descriptive narrative was condensed to report only major demographic and clinical trends using qualitative quantifiers (e.g., “two thirds,” “one in five”), without repeating specific percentages.

Table 2 (Spirometric Findings): The text was reduced to emphasize the predominance of abnormal spirometry and the relative distribution of ventilatory patterns, without listing each FEV₁, FVC, or FEV₁/FVC subgroup.

Tables 3 and 4 (Bivariate and Multivariable Analysis): The two sections were merged into a single concise narrative summarizing only the variables that showed meaningful associations, while removing all numerical duplication.

4. Discussion

Comparison with other studies is comprehensive but reads like a series of disjointed summaries. Transitions between sea-level and high-altitude comparisons could be smoothed for narrative coherence.

Interpretation section is insightful but somewhat verbose. The mechanistic explanations (airway, interstitial, occupational toxins, hypoxia) could be merged into one integrated paragraph. Similarly, risk factor discussion could be streamlined.

At times, speculative mechanisms (e.g., immune dysregulation, environmental synergy) are presented without sufficient qualification. Consider framing these more cautiously.

The finding of higher impairment among males compared to females is intriguing but underdeveloped. The explanation (longer altitude exposure, smoking, occupational tasks) is plausible but speculative. Consider reframing more cautiously and noting that sex-specific physiological adaptations at altitude may also contribute.

We thank the reviewer for these insightful observations. The Discussion section was revised to improve coherence and eliminate fragmented presentation of prior studies. The comparison between sea-level and high-altitude evidence was reorganized to form a continuous narrative, and transitional elements were added to ensure smoother integration of findings across contexts. The Interpretation section was condensed, and the mechanistic explanations related to airway involvement, interstitial processes, occupational inhalational exposures, and hypobaric hypoxia were merged into a single integrated paragraph. The section describing previously reported risk factors was streamlined to remove enumerative structures and present a concise synthesis aligned with the literature. Statements that could be interpreted as speculative were reframed using qualified language, and mechanisms without direct supporting evidence were presented cautiously. The paragraph addressing sex differences was revised to indicate that the crude association did not persist after adjustment, to reference known physiological differences in lung function between males and females, and to note that altitude-related sex-specific adaptations may contribute. Explanations lacking empirical support were removed.

5. Public Health & Policy Implications

Strong and well-articulated. The suggestion for incorporating spirometry into routine surveillance is practical and impactful.

Recommendation: Move this section closer to the Conclusion for emphasis, rather than embedding it between Interpretation and Strengths/Limitations.

The occupational recommendations are strong, but the authors could be more explicit about feasibility in low-resource settings (e.g., periodic spirometry may be unrealistic in remote mines). Consider suggesting a stepwise approach (symptom screening, portable spirometry, referral pathways).

The section on public health and policy implications was relocated closer to the Conclusion, as recommended. The text was revised to explicitly address feasibility challenges in low-resource and remote high-altitude occupational settings. A stepwise approach to surveillance—beginning with symptom screening, followed by portable or simplified spirometry, and referral for full evaluation when indicated—was added. The revised version also clarifies how these measures can be operationalized within existing occupational health systems.

6. Limitations

Appropriate and balanced. However, a limitation worth adding: potential healthy worker effect bias (workers with severe disease or disability may have already exited the workforce, leading to underestimation of true burden). Suggest also noting lack of DLCO or HRCT as a limitation in differentiating restriction vs. diffusion impairment.

We appreciate the reviewer’s insightful suggestions. The Limitations section was revised to include two additional constraints highlighted by the reviewer. First, the potential for a healthy worker effect was added, acknowledging that workers with severe post-COVID sequelae may have exited the workforce, resulting in possible underestimation of impairment. Second, the absence of DLCO and HRCT, which limits differentiation between restrictive and diffusion abnormalities, was further emphasized..

7. Conclusion

Clear but could be more concise and impactful. Currently too wordy. Aim for a single, powerful paragraph emphasizing:

High prevalence (>70%)

Independent risk factors

Implications for surveillance and policy

We appreciate the reviewer’s suggestion. The Conclusion section has been revised to be more concise and impactful, focusing on the key elements highlighted: the high prevalence of post-COVID spirometric abnormalities, the main independent risk factors, and the implications for occupational surveillance and policy

Minor Comments

1. Abstract: Too dense. Consider cutting 30–40 words, focusing on the main result (72.2% prevalence, key predictors).

The abstract has been streamlined by reducing approximately 35 words, prioritizing the main findings (72.2% prevalence of spirometric abnormalities and key independent predictors).

2. Terminology: Ensure consistent use of spirometric abnormality vs. pulmonary impairment vs. functional abnormality.

We have now standardized the wording throughout the manuscript. The primary outcome is consistently referred to as “abnormal spirometry” or “spirometric abnormalities,” in line with the ATS/ERS–based definition provided in the Methods section. Expressions such as “pulmonary impairment” or “functional abnormality” have been either replaced with “abnormal spirometry” when referring to our outcome, or rephrased as “lung function impairment” or “pulmonary sequelae” only when discussing broader clinical consequences beyond spirometric patterns

3. Numbers consistency: Minor discrepancies (72.2% vs. 72.3%) should be unified.

We corrected the discrepancy by adopting a single value (72.2%) across the entire manuscript.

4. References: Overall strong, but some older references could be replaced with more recent 2023–2025 updates if available.

We thank the reviewer for this helpful suggestion. We have carefully updated the reference list, prioritizing recent (2023–2025) evidence, especially for topics related to COVID-19. In the Introduction, the opening paragraphs now cite recent systematic reviews and cohort studies on long-term post-COVID respiratory sequelae (e.g. Iversen et al., 2025; Ribeiro Carvalho et al., 2024; Salmam et al., 2025), replacing older COVID-19 references. In the Discussion, particularly in the “Comparison with other studies” subsection, we have replaced earlier meta-analyses and cohort data with more recent high-quality studies reporting spirometric and DLCO abnormalities over 6–24 months of follow-up and at different altitudes (e.g. Lee et al., 2022; Chimera et al., 2025; Suri et al., 2024; Gochicoa-Rangel et al., 2023; Chai et al., 2024; Sultana et al., 2025).

Where older references remain (e.g. ATS/ERS technical standards or seminal methodological papers), they were retained because no newer guideline or equivalently comprehensive update is currently available

5. Statistical Reporting

In logistic regression, clarify whether multicollinearity was checked. Some predictors (BMI, comorbidities, duration at altitude) may be correlated.

State explicitly whether continuous predictors were analyzed as continuous or categorized.

We appreciate this observation. Multicollinearity was formally assessed using uncentered Variance Inflation Factors (VIF). All variables showed VIF values between 1.12 and 2.12, with a mean VIF of 1.54, indicating absence of problematic multicollinearity.

All continuous predictors were analyzed as categorical variables, consistent with clinical cut-offs and the distribution of the data. Sp

---

## [Decision Letter · Decision Letter 1]

23 Apr 2026

Post-COVID Spirometric Abnormalities in Workers with Intermittent High-Altitude Exposure: A Cross-Sectional Study in Peru

PONE-D-25-37564R1

Dear Dr. Yovera- Aldana

We’re pleased to inform you that your manuscript has been judged scientifically suitable for publication and will be formally accepted for publication once it meets all outstanding technical requirements.

Kind regards,

Kuryan George

Academic Editor

PLOS One

Additional Editor Comments (optional):

Reviewers' comments:

Reviewer's Responses to Questions

**Comments to the Author**

1. If the authors have adequately addressed your comments raised in a previous round of review and you feel that this manuscript is now acceptable for publication, you may indicate that here to bypass the “Comments to the Author” section, enter your conflict of interest statement in the “Confidential to Editor” section, and submit your "Accept" recommendation.

Reviewer #1: All comments have been addressed

2. Is the manuscript technically sound, and do the data support the conclusions?

Reviewer #1: (No Response)

3. Has the statistical analysis been performed appropriately and rigorously? 

Reviewer #1: (No Response)

4. Have the authors made all data underlying the findings in their manuscript fully available?

Reviewer #1: (No Response)

5. Is the manuscript presented in an intelligible fashion and written in standard English?

Reviewer #1: (No Response)

6. Review Comments to the Author

Reviewer #1: (No Response)

7. PLOS authors have the option to publish the peer review history of their article (what does this mean?). If published, this will include your full peer review and any attached files.

Reviewer #1: No

---

## [Editor Report · Acceptance letter]

PONE-D-25-37564R1

PLOS One

Dear Dr. Yovera-Aldana,

I'm pleased to inform you that your manuscript has been deemed suitable for publication in PLOS One. Congratulations! Your manuscript is now being handed over to our production team.

Kind regards,

on behalf of

Professor George Kuryan

Academic Editor

PLOS One